# 3D Magnetization Textures: Toroidal Magnetic Hopfion Stability in Cylindrical Samples

**DOI:** 10.3390/nano14010125

**Published:** 2024-01-04

**Authors:** Konstantin Guslienko

**Affiliations:** 1Departamento de Polímeros y Materiales Avanzados: Física, Química y Tecnología, Universidad del País Vasco, UPV/EHU, 20018 San Sebastián, Spain; kostyantyn.gusliyenko@ehu.eus; 2EHU Quantum Center, University of the Basque Country, UPV/EHU, 48940 Leioa, Spain; 3IKERBASQUE, The Basque Foundation for Science, 48009 Bilbao, Spain

**Keywords:** ferromagnetic materials, nanodots, nanowires, magnetization textures, topological charge, magnetic hopfion

## Abstract

Topologically non-trivial magnetization configurations in ferromagnetic materials on the nanoscale, such as hopfions, skyrmions, and vortices, have attracted considerable attention of researchers during the last few years. In this article, by applying the theory of micromagnetism, I demonstrate that the toroidal hopfion magnetization configuration is a metastable state of a thick cylindrical ferromagnetic nanodot or a nanowire of a finite radius. The existence of this state is a result of the competition among exchange, magnetostatic, and magnetic anisotropy energies. The Dzyaloshinskii–Moriya exchange interaction and surface magnetic anisotropy are of second importance for the hopfion stabilization. The toroidal hopfion metastable magnetization configuration may be reached in the process of remagnetizing the sample by applying an external magnetic field along the cylindrical axis.

## 1. Introduction

Topologically non-trivial two-dimensional (2D) and three-dimensional (3D) magnetization configurations in ferromagnetic materials, such as hopfions, skyrmions, vortices and domain walls, have attracted considerable attention from researchers during the last few years [1]. The stability of 3D magnetization configurations and the role of 3D (Hopf index), 2D (skyrmion number) topological charges and gyrovector in their dynamics are still far from complete understanding. Nowadays, 3D magnetization configurations in ferromagnetic materials on the nanoscale can be observed experimentally using electron holography or X-ray magnetic circular dichroism [2,3,4]. Very recently, the magnetic hopfions forming coupled states with skyrmion strings in FeGe submicron plates were observed using transmission electron microscopy [5].

The important question is the stability of the different 3D magnetization configurations. The standard approach to consider the stability and dynamics of magnetization configurations in magnetically ordered media is micromagnetism and the Landau–Lifshitz equation of motion of the magnetization field. It was established in the field theory [6,7] that any physical system with the second spatial derivatives in the Lagrange–Euler equation or squared gradient field term in the Lagrangian has no stable, time-independent, localized solutions in a 3D case for any form of the potential. Now, this statement is referred to as the Hobart—Derrick theorem. However, stable localized solutions (localized solitons) can exist if there are any energy contributions linear with respect to spatial derivatives or with higher-order spatial derivatives of the field [8,9]. A prominent example of the energy terms with the first derivatives is the so-called Lifshitz invariants, accounting for Dzyaloshinskii–Moriya interactions (DMIs) in magnetic materials with broken inversion symmetry. It was theoretically proved that such terms stabilize quasi-two-dimensional localized structures in the form of magnetic skyrmions [10]. Another opportunity to get stable 3D field configurations is accounting for the higher-order spatial derivatives in the Lagrangian. The terms quartic in spatial derivatives were suggested by Skyrme [11] and Faddeev [12] within the classical field theory. It was shown that the Faddeev–Skyrme Lagrangian has stable 3D localized soliton solutions in the form of toroidal hopfions [13,14]. The toroidal hopfions are some kinds of localized topological solitons and are characterized by non-zero values of 3D topological charge (Hopf index) [15,16]. It was recently shown [17] that the classical Heisenberg model with competing long-range exchange interactions can result in quadratic terms in the second spatial derivatives of the magnetization field. Although such a model is beyond the standard theory of micromagnetism, it may result in the stabilization of toroidal magnetic hopfions. The question is whether it is possible to stabilize magnetic hopfions in a ferromagnet within the standard exchange approach (avoiding exotic exchange interactions) due to non-zero DMI terms and/or magnetostatic energy. Such energy contributions are beyond the field theory, and the applicability of the Hobart–Derrick theory to the evaluation of the stability of magnetic field configurations should be reconsidered. A simple scaling analysis accounting for DMI terms was conducted in Ref. [18]. However, this analysis ignored the magnetostatic interaction (which is unavoidable in real ferromagnetic samples) and the finite sample sizes. The non-local magnetostatic interaction is usually not considered in the theory of magnetic skyrmions and hopfions or is accounted for in a simplified form. Skyrmions are considered either in the bulk magnetic crystals without inversion symmetry or in ultrathin films. In both cases, the magnetostatic interaction is reduced to a local form of some extra contribution to the magnetic anisotropy energy. Accounting for the magnetostatic interaction in relatively thick magnetic dots [19] allows for stabilizing quasi-2D skyrmions without the presence of any DMI if a small out-of-plane magnetic anisotropy is included in the energy functional. The magnetostatic interaction was not included in the energy functional in Refs. [20,21,22], describing magnetic hopfions in thick cylindrical dots, without explanations of its importance. I note that the magnetostatic interaction can also lead to the stabilization of other kinds of complicated magnetization textures: the Bloch point hopfions with non-zero Hopf index or half-hedgehog (3D quasi-skyrmion) magnetization textures, even in soft magnetic materials with no DMI [23,24].

It is expected that, in the process of the toroidal hopfion translation motion, the gyroforce perpendicular to the hopfion velocity is absent due to the nullification of the global hopfion gyrovector [22]. The hopfion motion induced by a spin-polarized current should be along the driving force direction (avoiding the undesirable skyrmion Hall effect). Therefore, toroidal hopfions would be used in racetrack data storage devices as information carriers [4,22].

In this article, I consider the magnetic energy functional consisting of exchange, magnetostatic, Dzyaloshinskii–Moriya and magnetic anisotropy energies. I show that magnetostatic energy is crucial for toroidal hopfion stability (metastability) in cylindrical ferromagnetic nanodots and nanowires. The DMI energy term also supports the stabilization of magnetic toroidal hopfions. However, the DMI energy is of secondary importance compared to the magnetostatic energy, and the hopfion can be stable in soft magnetic materials with no DMI.

## 2. Materials and Methods

In this section, I provide a definition of the 3D topological charge (Hopf index) of a magnetization texture and present explicit equations describing the magnetization of the toroidal magnetic hopfion. Then, I analyze the energy and stability of the magnetic hopfion in magnetic cylindrical nanodots and nanowires using the methods of the theory of micromagnetism and determine the main magnetic and geometrical parameters necessary for the existence of stable hopfion configurations in the restricted cylindrical geometry.

The Hopf index of a 3D magnetization configuration is calculated as a volume integral from the dot product of the emergent magnetic field vector potential and the emergent magnetic field A·B [16,25]. We define the unit magnetization vector $m\left(r\right)=M\left(r\right)/M_{s}$, where $M_{s}$ is the material saturation magnetization. The Hopf invariant for the mapping of the 3D coordinate space (**r**) to the unit sphere $m{\left(r\right)}^{2}=1$ in the magnetization space is
(1)$$Q_{H}=\frac{1}{{\left(4\pi \right)}^{2}}\int dVA\cdot B.$$

An inhomogeneous spin texture $m\left(r\right)$ allows us to define the emergent electromagnetic field tensor $F_{\mu \nu }\left(m\right)=m\cdot \left(\partial_{\mu }m\times \partial_{\nu }m\right)$, where $\partial_{\mu }=\partial /\partial x_{\mu }$ denotes derivatives with respect to the components of the 3D radius-vector ***r***. Similar to classical electrodynamics, the emergent field tensor and emergent magnetic field can be expressed via the vector potential A as $F_{\mu \nu }\left(m\right)=\partial_{\mu }A_{\nu }\left(m\right)-\partial_{\nu }A_{\mu }\left(m\right)$ and B=∇×A [26,27].

A given magnetization texture $m\left(r\right)$ leads to an unambiguous expression for the emergent magnetic field B components $B_{\lambda }\left(m\right)=\varepsilon_{\lambda \mu \nu }F_{\mu \nu }\left(m\right)/2$. However, to find the Hopf index (1), one needs to construct the emergent field vector potential A, as shown recently in Ref. [27]. There is another approach to calculate the Hopf index and explicitly find the hopfion magnetization $m\left(r\right)$ configuration in infinite media [27]. The approach is immediately based on the Hopf mapping. The Hopf mapping of the 3D coordinate space $R^{3}$ to the unit sphere *S*_2_(***m***) in the magnetization space is $m=Z^{+}\sigma Z$ [28], where $\sigma =\left(\sigma_{x},\sigma_{y},\sigma_{z}\right)$ are the Pauli matrices, and $Z={\left(Z_{1},Z_{2}\right)}^{T}$ is a spinor composed of the coordinates $X_{i}$ (*i* = 1, 2, 3, 4) of the unit radius hypersphere $S_{3}\left(X\right)$ in 4D space. The spinor components used in the definition of the Hopf mapping are $Z_{2}=X_{1}+iX_{2}$, $Z_{1}=X_{4}+iX_{3}$. It was shown in Ref. [27] that there is a simple relation between the magnetization m and spinor Z components. The Hopf index (3D topological charge) can then be represented as the triple dot product of the gradients of the spinors Z:(2)$$Q_{H}=\frac{1}{{\left(4\pi \right)}^{2}}\int dVZ^{+}\nabla Z\cdot \left(\nabla Z^{+}\times \nabla Z\right).$$
This form is similar to the skyrmion number (2D topological charge).

The expressions for the hopfion magnetization $m\left(r\right)$ components are simplest in toroidal coordinates $r\left(\eta ,\beta ,\varphi \right)$ [13,14]. Below, we calculate the magnetic hopfion energy in ferromagnetic circular cylindrical samples. Therefore, it is convenient to use cylindrical coordinates $r\left(\rho ,\varphi ,z\right)$ of the radius vector r. The cylindrical $\left(\rho ,\varphi ,z\right)$ and toroidal $\left(\eta ,\beta ,\varphi \right)$ coordinates are related as follows: $\rho =asinh\left(\eta \right)/\tau$, $z=asin\left(\beta \right)/\tau$, φ=φ, and $\tau =cosh\left(\eta \right)-cos\left(\beta \right)$, where the toroidal parameter η varies from 0 to ∞, the poloidal angle β varies from −π to π, the azimuthal angle φ varies from 0 to 2π [29], and a is a geometrical scale parameter having the sense of the hopfion radius. According to Ref. [27], the out-of-plane *z*-component of the hopfion magnetization depends only on the toroidal parameter η and can be explicitly expressed as
(3)$$m_{z}\left(\eta \right)=p\frac{1-cosh^{2m}\left(\eta \right)tanh^{2n}\left(\eta \right)}{1+cosh^{2m}\left(\eta \right)tanh^{2n}\left(\eta \right)},$$
where *m* and *n* are integer numbers, $p=m_{z}\left(\eta =0\right)$ is the hopfion polarity, p=±1.

The hopfion in-plane magnetization $m\left(r\right)$ components in toroidal coordinates are determined using the expression [27]
(4)$$m_{x}\left(r\right)+im_{y}\left(r\right)=\sqrt{1-m_{z}^{2}\left(\eta \right)}exp\left[i\left(n\varphi +m\beta \right)\right].$$
This involves the poloidal and azimuthal angles β and φ and the winding numbers (*m* and *n*) in the poloidal and azimuthal directions, respectively.

By substituting the hopfion magnetization components given using Equations (3) and (4) into the definition of the Hopf index in Equation (1) or Equation (2), we can find the Hopf index of the hopfion magnetization texture. The calculated Hopf index for the toroidal magnetic hopfion is $Q_{H}=mnp$, i.e., it is an integer for an infinite sample and is proportional to the product of the poloidal *m* and azimuthal *n* winding numbers. For our calculations on the magnetic hopfion energy and stability, we consider the simplest hopfion with n=1, m=1 and $\left|Q_{H}\right|\;$= 1, which is assumed to be the lowest energy toroidal hopfion in cylindrical geometry. For this particular case, the z-component of the hopfion magnetization $m_{z}\left(\eta \right)$ given using Equation (3) is essentially simplified to $m_{z}\left(\eta \right)=p\left(1-2tanh^{2}\left(\eta \right)\right)$.

The physical system under consideration is a thick cylindrical ferromagnetic dot or cylindrical nanowire of radius *R* and thickness (length) *L*. The limit of an infinite cylinder R→∞ and L→∞ is assumed because the hopfion magnetization given using Equations (3) and (4) is derived for infinite space. Although the toroidal hopfion magnetization field (3) and (4) has the simplest representation in toroidal coordinates, to calculate the hopfion energy in a cylindrical sample, we need to change the toroidal coordinates to the cylindrical ones. The energy functional consists of contributions from the exchange, DMI, uniaxial anisotropy and magnetostatic energies:(5)$$E\left[m\right]=\int d^{3}r\left[A\sum_{\alpha }{\left(\nabla m_{\alpha }\right)}^{2}+D\left(m\cdot rotm\right)+K\left(1-m_{z}^{2}\right)-\frac{1}{2}M_{s}m\cdot H_{m}\left(r\right)\right],$$
where *A* is the exchange stiffness constant, *D* is the DMI parameter, and the magnetostatic field $H_{m}\left(r\right)$ is calculated within the magnetostatic Green function formalism [30], $H_{m}\left(r\right)=M_{s}\int d^{3}r^{\prime }\hat{G}\left(r,r^{\prime }\right)m\left(r^{\prime }\right)$, ${\left(\hat{G}\left(r,r^{\prime }\right)\right)}_{\alpha \beta }=-\partial^{2}/\partial x_{\alpha }\partial x_{\beta }\left(1/\left|r-r^{\prime }\right|\right)$.

The idea is to calculate the magnetic energy functional $E\left[m\right]$ given using Equation (5) as a function of the hopfion radius $E\left(a\right)$, substituting the toroidal hopfion magnetization given using Equations (3) and (4) to $E\left[m\right]$. To calculate the different contributions to the magnetic energy (5), we represent the hopfion magnetization $m\left(r\right)$ components via the spherical angles Θ and Φ: $m_{z}=cos\Theta$, $m_{x}+im_{y}=sin\Theta exp\left(i\Phi \right)$, and use the cylindrical coordinates $r\left(\rho ,\varphi ,z\right)$ for the radius-vector ***r***. The magnetization spherical angles are functions of the position within the sample ***r***, $\Theta =\Theta \left(r\right)$ and $\Phi =\Phi \left(r\right)$. Following the theory of 2D magnetic solitons (vortices and skyrmions) [31], we choose the hopfion magnetization spherical angles in axially symmetric form: $\Theta \left(r\right)=\Theta \left(\rho ,z\right)$ and $\Phi \left(r\right)=n\varphi +\gamma \left(\rho ,z\right)$. The angle $\gamma \left(\rho ,z\right)=m\beta \left(\rho ,z\right)$ is the variable hopfion helicity. The in-plane hopfion magnetization components are $m_{\rho }=\sin\Theta \cos\gamma$ and $m_{\varphi }=\sin\Theta \sin\gamma$. The explicit form of the functions $\eta \left(\rho ,z\right)$ and $\beta \left(\rho ,z\right)$ is given by the expressions $\eta \left(\rho ,z\right)=atanh\left(2a\rho /\left(\rho^{2}+z^{2}+a^{2}\right)\right)$ and $\beta \left(\rho ,z\right)=atan\left(2az/\left(\rho^{2}+z^{2}-a^{2}\right)\right)$. These expressions allow us to rewrite the hopfion magnetization (3) for *p* = +1 in the cylindrical coordinates $\left(\rho ,\varphi ,z\right)$ as $m_{z}\left(\rho ,z\right)=1-8\rho^{2}a^{2}/{\left(a^{2}+\rho^{2}+z^{2}\right)}^{2}$. The toroidal hopfions can be approximately interpreted as twisted skyrmion strings with their centers located in the *xOy* plane (*z* = 0) and described using the equation ρ=a. The poloidal angle β describes the twist angle around the ring ρ=a. However, there is an important difference between the twisted skyrmion string and the hopfion magnetization configuration. The string magnetization in its center is directed along the unit vector of the cylindrical coordinate system $\hat{\rho }$ ($m=\pm \hat{\rho })$ in the *xOy* plane, whereas, for the toroidal hopfion, the magnetization m at the ring ρ=a (η=∞) is directed along the *z*-axis $m=-p\hat{z}$, oppositely to the magnetization in the hopfion center ρ=0. The iso-surface $m_{z}\left(\eta \right)=const$ of the toroidal hopfion is schematically shown in Figure 1.

We express the spatial coordinates $\left(\rho ,z\right)$ in the units of the hopfion radius a, which has the sense of the scale parameter. Then, by substituting the hopfion magnetization $m\left(r\right)$ components (3) and (4) into the energy (5) and accounting for the expression $\left(m\cdot rotm\right)=-sin^{2}\Theta \partial \gamma /\partial z$ for the even on out-of-plane coordinate *z* part of the DMI energy, one can find explicitly the local exchange and DMI energies for an infinite sample in the following form:(6)$$E_{ex}=32\pi^{2}Aa,\;E_{D}=-64\pi DI_{d}a^{2},$$
where $I_{d}$ is some integral, which was evaluated numerically to be $I_{d}=0.26$ for the hopfion angles $\Theta \left(\rho ,z\right)$ and $\gamma \left(\rho ,z\right)$.

The uniaxial magnetic anisotropy contribution $E_{a}=2\pi K\int d\rho \rho \int dzsin^{2}\Theta$ and the magnetostatic contribution diverge as the sample radius *R* increases, approximately as ~lnR (where *R* is the upper limit of the integration over the polar radius vector *ρ*). Although the toroidal hopfion is a localized soliton, and the far-field magnetization is asymptotically trivial $m\left(r\right)\to m_{0}$ (uniform magnetization background) at $\left|r\right|\to \infty$, the degree of the soliton localization in the radial ρ-direction is not sufficient to obtain finite anisotropy and magnetostatic energies at R→∞.

Therefore, we use a finite in-plane sample size *R* when calculating the energy contributions defined through Equation (5) using the toroidal hopfion magnetization given in Equations (3) and (4). Although these equations describe the hopfion in an infinite sample, we use them below as trial functions to find the hopfion magnetic energy in a finite cylindrical sample of radius *R*, assuming that *R* is large enough. The cylinder thickness (wire length) *L* can be finite or infinite.

The magnetic anisotropy term can be written as
(7)$$E_{a}=4\pi^{2}M_{s}^{2}QI_{a}\left(\frac{R}{a}\right)a^{3},$$
where $Q=K/2\pi M_{s}^{2}$, $I_{a}\left(x\right)=2\int_{0}^{x}d\rho \rho \int_{0}^{z_{m}}dzsin^{2}\Theta$, $z_{m}=\beta x/2$ and β=L/R.

To calculate the components of the Green’s function tensor, we use the Coulomb kernel $1/\left|r-r^{\prime }\right|$ decomposition via the Bessel functions of the first kind $J_{\mu }\left(x\right)$:$$\frac{1}{\left|r-r^{\prime }\right|}=\int_{0}^{\infty }dkexp\left(-k\left|z-z^{\prime }\right|\right)\sum_{\mu =-\infty }^{\infty }J_{\mu }\left(k\rho \right)J_{\mu }\left(k\rho^{\prime }\right)\exp\left[i\mu \left(\varphi -\varphi^{\prime }\right)\right].$$

The hopfion magnetization $m\left(r\right)$ components in the cylindrical coordinates do not depend on the azimuthal angle φ. Therefore, we can average the dipolar field $H_{m}\left(r\right)$ over φ. This leads to the axially symmetric field in the form $H_{m}\left(r\right)=\left(H_{m}^{\rho }\left(\rho ,z\right),0,H_{m}^{z}\left(\rho ,z\right)\right).$ The magnetostatic field is related to the magnetization components via the averaged Green’s functions $g_{\alpha \beta }\left(\rho ,\rho^{\prime },z,z^{\prime }\right)=\int_{0}^{2\pi }d\varphi \int_{0}^{2\pi }d\varphi^{\prime }\;{\left(\hat{G}\left(r,r^{\prime }\right)\right)}_{\alpha \beta }/2\pi$. The components $g_{\alpha \beta }$ are equal to zero if at least one of the indices α   or β is equal to φ. Only the components $g_{\rho \rho },g_{\rho z},\;\;g_{z\rho }\;\mathrm{and}\;g_{zz}$ are not equal to zero. The contribution of the components $g_{\rho z}\;\mathrm{and}\;g_{z\rho }$ to the magnetostatic energy disappears due to the system cylindrical symmetry and the hopfion's axial symmetry. Therefore, the nonlocal magnetostatic energy can be written as
(8)$$E_{m}=-\pi M_{s}^{2}\int d\rho \rho \int dz\int d\rho^{\prime }\rho^{\prime }\int dz^{\prime }\left[g_{\rho \rho }m_{\rho }\left(\rho ,z\right)m_{\rho }\left(\rho^{\prime },z^{\prime }\right)+g_{zz}m_{z}\left(\rho ,z\right)m_{z}\left(\rho^{\prime },z^{\prime }\right)\right].$$

The hopfion magnetostatic energy (8) is the sum of two contributions, ρρ and zz, which depend on the hopfion magnetization components $m_{\rho }$ and $m_{z}$, respectively. The first contribution can be expressed as
(9)$$E_{m}^{\rho \rho }=2\pi^{2}M_{s}^{2}I_{m}^{\rho }\left(\frac{R}{a}\right)a^{3},$$
where the integral $I_{m}^{\rho }\left(x\right)$ is defined as $I_{m}^{\rho }\left(x\right)=\int dz\int dz\text{’}I_{\rho }\left(z,k,x\right)\;I_{\rho }\left(z\text{’},k,x\right)$, $I_{\rho }\left(z,k,x\right)=-k\int_{0}^{x}d\rho \rho J_{1}\left(k\rho \right)m_{\rho }\left(\rho ,z\right)$, and $m_{\rho }\left(\rho .z\right)=sin\;\Theta \left(\rho ,z\right)cos\;\gamma \left(\rho ,z\right)$. Here, $J_{1}\left(x\right)$ is the Bessel function of the first kind. The upper and lower limits in the integral over the thickness coordinates z and z’ are ±∞ for magnetic wires. However, the limits are finite and equal ±βx/2 for the magnetic dots or finite length wires with the aspect ratio β=L/R.

The integral in Equation (8) cannot be calculated analytically/numerically and is too complicated to operate with. We note that the function $exp\left(-k\left|z-z^{\prime }\right|\right)$ in the definition of $I_{m}^{\rho }\left(x\right)$ has a sharp maximum at $z=z^{\prime }$; therefore, we can substitute $I_{\rho }\left(z^{\prime },k,x\right)$ for $I_{\rho }\left(z,k,x\right)$ under the integral sign. Then, the integral is essentially simplified to be $I_{m}^{\rho }\left(x\right)=2\int_{-z_{m}}^{z_{m}}dz\int_{0}^{\infty }dkk^{-1}\left[1-exp\left(-kz_{m}\right)\cos h\left(kz\right)\right]{\left(I_{\rho }\left(z,k,x\right)\right)}^{2}$, $z_{m}=\beta x/2$ It can be shown that within the limit β≫1 (cylindrical wire), the integral is reduced to the simple expression, $I_{m}^{\rho }\left(x\right)=4\int_{0}^{\beta x/2}dz\int_{0}^{x}d\rho \rho {\left(m_{\rho }\left(\rho ,z\right)\right)}^{2}$. It has the form of an effective hard axis magnetic anisotropy in the ρ-direction, normal to the cylinder side surface. This anisotropy is analogous to the shape anisotropy of a uniformly magnetized wire along its length.

The second, zz contribution to the magnetostatic energy can be presented in a form similar to Equation (9):(10)$$E_{m}^{zz}=2\pi^{2}M_{s}^{2}I_{m}^{z}\left(\frac{R}{a}\right)a^{3},$$
where, in the local approximation, the integral $I_{m}^{z}\left(x\right)=2\int_{-z_{m}}^{z_{m}}dz\int_{0}^{\infty }dkkexp\left(-kz_{m}\right)\cos h\left(kz\right){\left(I_{z}\left(z,k,x\right)\right)}^{2}$ and $I_{z}\left(z,k,x\right)=\int_{0}^{x}d\rho \rho J_{0}\left(k\rho \right)m_{z}\left(\rho ,z\right)$.

Using the energy contributions described in Equations (6)–(10), we can write the total dimensionless magnetic energy $\varepsilon \left[m\right]=E\left[m\right]/\left(4\pi M_{s}^{2}l_{e}^{3}\right)$ of the toroidal hopfion in the cylindrical dot/wire in the units of $4\pi M_{s}^{2}l_{e}^{3}$,
(11)$$\varepsilon \left(a\right)=16\pi^{2}a-16dI_{d}a^{2}+\pi QI_{a}\left(\frac{R}{a}\right)a^{3}+\frac{\pi }{2}I_{m}^{\rho }\left(\frac{R}{a}\right)a^{3}+\frac{\pi }{2}I_{m}^{z}\left(\frac{R}{a}\right)a^{3},$$
where $l_{e}=\sqrt{A/2\pi M_{s}^{2}}$ is the material exchange length, $d=D/M_{s}^{2}l_{e}$ is the reduced DMI parameter, and the hopfion radius *a* is presented in the units of $l_{e}$.

Although the exchange and DMI energies in Equation (11) have a finite limit at R→∞, we need to rewrite them for finite values of the ratio R/a, similar to the magnetic anisotropy and magnetostatic energies. We use the expressions
(12)$$\varepsilon_{ex}\left(a\right)=\pi aI_{ex}\left(\frac{R}{a}\right),\;\varepsilon_{D}\left(a\right)=-\frac{1}{2}da^{2}I_{D}\left(\frac{R}{a}\right),$$
where $I_{ex}\left(x\right)=\int_{0}^{x}d\rho \rho \int_{-z_{m}}^{z_{m}}dz\left[{\left(\nabla m_{z}\right)}^{2}/sin^{2}\Theta +sin^{2}\Theta \left\{{\left(\nabla \gamma \right)}^{2}+1/\rho^{2}\right\}\right]$, and $I_{D}\left(x\right)=\int_{0}^{x}d\rho \rho \int_{-z_{m}}^{z_{m}}dzsin^{2}\Theta \partial \gamma /\partial z$. The limiting values are $I_{ex}\left(x\to \infty \right)=16\pi$ and $I_{D}\left(x\to \infty \right)=32I_{d}$, in agreement with Equation (6).

Generalizing Equation (11) for a finite value of the cylindrical dot/wire radius *R* and arbitrary value *L* of the dot thickness (wire length), the total normalized magnetic energy of the toroidal hopfion is given by
(13)$$\varepsilon \left(a,R\right)=\pi aI_{ex}\left(\frac{R}{a}\right)-\frac{1}{2}da^{2}I_{D}\left(\frac{R}{a}\right)+\pi QI_{a}\left(\frac{R}{a}\right)a^{3}+\frac{\pi }{2}I_{m}^{\rho }\left(\frac{R}{a}\right)a^{3}+\frac{\pi }{2}I_{m}^{z}\left(\frac{R}{a}\right)a^{3}.$$

We note that the hopfion energy (13) is essentially more complicated than the simple scaling polynomial equations with respect to the hopfion radius a used in Ref. [18] due to the presence of the magnetostatic interaction and finite system size *R*. The magnetic anisotropy and magnetostatic terms should be considered at a finite value of *R* due to their divergence at R→∞. The hopfion energy depends not only on the scale parameter (hopfion radius) a but also on the cylindrical sample radius *R* and the aspect ratio β=L/R.

## 3. Results and Discussion

The reduced hopfion magnetic energy (13) is valid for any ferromagnetic material with uniaxial magnetic anisotropy (the parameter Q) and Dzyaloshinskii–Moria exchange interaction (the parameter d) for the given cylindrical sample sizes, R and L. The material exchange length $l_{e}$ serves as a natural scale for the hopfion radius a and the dot/wire radius R. The hopfion magnetic energy $\varepsilon \left(a,R\right)$ (13) vs. the hopfion radius a is plotted in Figure 2, Figure 3 and Figure 4 for the cylinder radii *R* = 100 nm and 250 nm. To plot Figure 2 and Figure 3, we used a set of magnetic material parameters: $l_{e}=$ 5 nm, d=1 and *K* = 0. Such a value of $l_{e}$ can be obtained, in particular, using A= 11 pJ/m and *M*_s_ = 837 kA/m, which are typical for soft magnetic materials such as Ni_80_Fe_20_ alloy (permalloy). However, we used finite values of the magnetic anisotropy constant *K* to plot Figure 4. There is a pronounced minimum of the magnetic energy at finite values of the hopfion radius a, which corresponds to the hopfion stable state.

The magnetostatic terms in Equation (13) are mainly responsible for the appearance of the minimum of the hopfion energy $\varepsilon \left(a,R\right)$ at $a_{0}\approx \left(0.90÷0.92\right)R$ in soft magnetic materials (K=0). The reduced equilibrium hopfion radius $a_{0}/R$ is approximately equal to 0.9 and weekly depends on the sample's magnetic and geometrical parameters. The magnetostatic energy contributions given in Equations (9) and (10) are functions only of x=R/a and β. Therefore, the reduced equilibrium value of $a_{0}/R$ depends only on β if only the magnetostatic energy is accounted. The weak dependence $a_{0}/R=f\left(\beta ,R,d\right)$ on other sample parameters reflects small contributions of the exchange and DMI energies to the total magnetic energy of the toroidal hopfion. However, the equilibrium hopfion radius $a_{0}$ depends on the magnetic anisotropy constant, especially at high values of the cylinder aspect ratio β.

The uniaxial magnetic anisotropy energy in Equation (13) strongly influences the hopfion stability and the value of the equilibrium hopfion radius. It renormalizes, in some sense, the magnetostatic contribution, which can be approximately treated as an effective hard axis magnetic anisotropy in the in-plane ρ-direction. The magnetic anisotropy destabilizes the hopfion state at K>0 (“easy axis” anisotropy) or stabilizes it for K<0 (“easy plane” anisotropy), as shown in Figure 4. The positive magnetic anisotropy energy at K>0 increases the influence of the positive magnetostatic energy contribution. The energy minimum becomes a shadow and disappears at large values of *K*. The negative “easy plane” anisotropy energy (K<0) competes with the magnetostatic energy. This leads to the negative magnetic hopfion energy and a deeper energy minimum at moderate values of $\left|K\right|$, as depicted in Figure 4. The “easy axis” (“easy plane”) magnetic anisotropy leads to a decrease (increase) in the equilibrium hopfion radius $a_{0}$.

It is reasonable to use a cylinder aspect ratio β≥2 in the calculations of the hopfion energy due to the strong localization of the hopfion in the *z*-direction. The DMI term, at a typical value of d~1 (D~1 mJ/m^2^), is essentially smaller than the magnetostatic term and results in a small modification of the hopfion magnetic energy. Thus, to stabilize the toroidal hopfion in a cylindrical dot/wire, we can ignore DMI and focus on the hopfion stabilization in strong ferromagnets. The hopfion energy value at the minimum $\varepsilon \left(a_{0}\right)$ in soft magnetic materials (*K* = 0) and magnetic materials with an “easy axis” (K>0) magnetic anisotropy is typically higher than the energy $\varepsilon \left(SD\right)$ of the out-of-plane single-domain (SD) state, especially at large values of β. Therefore, the toroidal hopfion is not the ground state of the cylindrical dot or wire with zero or positive uniaxial magnetic anisotropy constant. The ground state is longitudinally magnetized dot/wire with an almost uniform magnetization configuration ($m_{z}=\pm 1$). In good approximation, the magnetic energy is given using $\varepsilon \left(SD\right)=\left(\pi /2\right){\left(R/l_{e}\right)}^{3}\beta \left[N_{zz}\left(\beta \right)-Q\right]$, where $N_{zz}\left(\beta \right)=2\beta^{-1}\int_{0}^{\infty }dkk^{-2}J_{1}^{2}\left(k\right)\left[1-exp\left(-\beta k\right)\right]$ is the cylinder demagnetizing factor along the axial *z*-direction [32], and $Q=K/2\pi M_{s}^{2}$. The SD state energy can be obtained from the hopfion energy given in Equation (13 de) within the limit a→0. The situation is drastically changed for the “easy plane” anisotropy K<0 (Figure 4). The hopfion energy $\varepsilon \left(a_{0}\right)$ can be negative at moderate values of $\left|K\right|$ (the value $\left|K\right|$ = 0.22 MJ/m^3^ was used to plot Figure 4), and, therefore, $\varepsilon \left(a_{0}\right)<\varepsilon \left(SD\right)$. However, there is no guarantee that the toroidal hopfion is the ground state of the cylindrical dot/wire with the “easy plane” anisotropy K<0 because other inhomogeneous 3D magnetic configurations may have lower energy than the hopfion energy $\varepsilon \left(a_{0}\right)$. An example of such an inhomogeneous configuration is the toron (consisting of two Bloch points) recently calculated in cylindrical [33] and square magnetic dots [34].

We note that there is an energy barrier between the hopfion magnetization state $\varepsilon \left(a_{0}\right)$ and the single-domain state as $\varepsilon \left(SD\right)$, a→0. The hopfion energy $\varepsilon \left(a\right)$ given in Equation (13) asymptotically approaches the value of $\varepsilon \left(SD\right)$ as the hopfion radius increases a/R→∞. Apparently, this limit also describes the cylindrical dot/wire single-domain state with the magnetization along the cylindrical dot/wire axis. The SD magnetization configuration limits, a→0 and a→∞, correspond to the hopfion collapse or infinite extension in the radial direction, correspondingly. The energy minimum at a finite value of $a=a_{0}$ is separated from the longitudinal SD state a→∞ through an energy barrier.

The calculation method presented above allows us to conclude about the toroidal hopfion energy $\varepsilon \left(a\right)$ as a function of the hopfion radius and identify the energy minimum at a specific value of the hopfion radius, $a=a_{0}$ (see Figure 2, Figure 3 and Figure 4). Except for a minimum at $a=a_{0}$, the hopfion energy vs. a reveals some maxima at $a=a_{m}^{\pm }$, where (+) represents increasing and (−) means decreasing value of a. The energy difference $\Delta E\left(a_{m}^{\pm }\right)=E\left(a_{m}^{\pm }\right)-E\left(a_{0}\right)$ can be treated as an energy barrier for transitions from the hopfion to the out-of-plane single-domain state (described as limiting cases a→0 and a→∞) within the toroidal hopfion model. However, the hopfion radius may not be an appropriate “reaction coordinate” to describe a transition from the hopfion to the out-of-plane single-domain magnetization configuration. A more realistic path in multidimensional parameter space can include some other intermediate magnetization configurations, which are very different from the toroidal magnetic hopfion. The saddle point along this path corresponds to an energy barrier, which can be essentially smaller than the barrier calculated within the approach of the rigid hopfion transformation mode. Using the ferromagnetic cylinder parameters as described in the caption to Figure 2, one can find the energy barriers as $\Delta E\left(a_{m}^{-}\right)/k_{B}T=$ 2.39 × 10^5^ for β=2 and $\Delta E\left(a_{m}^{-}\right)/k_{B}T=$ 1.97 × 10^5^ for β=8 ($R=20l_{e}=100$ nm). The energy scale is $4\pi M_{s}^{2}l_{e}^{3}=$ 1.10 × 10^−19^ J. The energy barriers are much bigger than the thermal energy $k_{B}T$ at room temperature *T* (assuming that *T* is much lower than the Curie temperature $T_{c}$, where $M_{s}\left(T_{c}\right)\to 0$ and the barriers disappear). Note that these huge energy barriers found within the toroidal hopfion model by varying the hopfion radius are overestimated. The calculation of more realistic energy barriers corresponding to a transient configuration for the transformation from the toroidal hopfion to the single-domain magnetization configuration is beyond the scope of the present article. These energy barriers are of interest for the hopfion thermostability on a long-time scale. The calculations within the toroidal hopfion model showed that the hopfion energy minimum is very deep, and the hopfion magnetization configuration is thermostable.

We note that for the finite cylindrical samples, another hopfion ansatz was suggested [20] and used in Ref. [22]. This ansatz is a good approximation to minimize the hopfion energy if the magnetostatic energy contribution is ignored and a strong surface out-of-plane magnetic anisotropy is introduced by enforcing the boundary conditions $m=\hat{z}$ at the dot's top/bottom faces z=±L/2. The magnetostatic interaction is numerically accounted for in Refs. [33,35]. However, the authors of these papers believe that the strong out-of-plane magnetic anisotropy (*K* = 0.8 MJ/m^3^ in the surface layers [35] or the surface anisotropy *K*_s_ = 0.5 mJ/m^2^ [33]) along with DMI are necessary for the hopfion stabilization in cylindrical dots or infinite films. Calculation of the surface magnetic anisotropy contribution to the hopfion energy (Equation 13) shows that it is negligible for the surface anisotropy values *K*_s_ of the order of 1 mJ/m^2^. We demonstrated in Figure 2, Figure 3 and Figure 4 that the main contribution to the hopfion energy comes from the magnetostatic interaction, which is unavoidable present for all inhomogeneous magnetization textures in the restricted geometry of cylindrical thick magnetic dots and wires. Although DMI and uniaxial out-of-plane surface magnetic anisotropy may be accounted for in the energy functional, they are of secondary importance for the toroidal hopfion stabilization.

The toroidal hopfion metastable magnetization configuration may be reached in the process of the sample remagnetizing by applying a magnetic field along the cylindrical axis *Oz* as an intermediate metastable state in the low-field part of the hysteresis loop, $\langle M_{z}\left(H_{z}\right)\rangle =V^{-1}\int dVM_{z}\left(r,H_{z}\right)$. Using the hopfion magnetization (3), we can find the volume-averaged reduced magnetization $\mu_{z}\left(a,\beta \right)=\langle m_{z}\left(\rho ,z\right)\rangle$ at zero magnetic field $H_{z}=0$, which has the sense of the hopfion remanent magnetization. The equilibrium remanent magnetization $\mu_{z}\left(a_{0},\beta \right)$ is an increasing function of the cylinder aspect ratio β, saturating at β≫1, $\mu_{z}\left(a_{0},\beta \gg 1\right)\to 1$. We note that at a fixed value of the cylinder aspect ratio β, the remanent magnetization $\mu_{z}\left(a,\beta \right)$ in soft magnetic materials has a minimum at the hopfion radius $a_{\mu }$ approximately equal to the hopfion equilibrium radius $a_{0}$, $a_{\mu }\approx a_{0}$, for any value of β≥2. An external magnetic field applied to the sample in any direction will suppress the inhomogeneous magnetization hopfion configuration and result in the saturated magnetization state at some critical value of the field. However, the hopfion hysteresis loop is beyond the scope of the present article and could be the subject of future investigations.

## 4. Conclusions

It is demonstrated that the toroidal hopfion magnetization configuration is a metastable state of a thick cylindrical ferromagnetic nanodot or nanowire with a finite radius *R*. The existence of this state in soft magnetic materials is a result of the competition between the exchange and magnetostatic energies. The reduced equilibrium hopfion radius $a_{0}/R$ is approximately equal to 0.9 and weekly depends on the sample's magnetic end geometrical parameters. The uniaxial “easy axis” magnetic anisotropy (anisotropy constant K>0) destabilizes the hopfion state, whereas the “easy plane” magnetic anisotropy (K<0) facilitates the hopfion stabilization. The Dzyaloshinskii–Moriya exchange interaction and the out-of-plane surface magnetic anisotropy are of secondary importance for hopfion stabilization. The toroidal hopfion metastable magnetization configuration may be reached during the process of the sample remagnetizing by applying an external magnetic field along the cylindrical axis. The hopfion magnetization configuration corresponds to a deep magnetic energy minimum and is stable with respect to thermal fluctuations.

## Figures and Tables

**Figure 1 nanomaterials-14-00125-f001:**
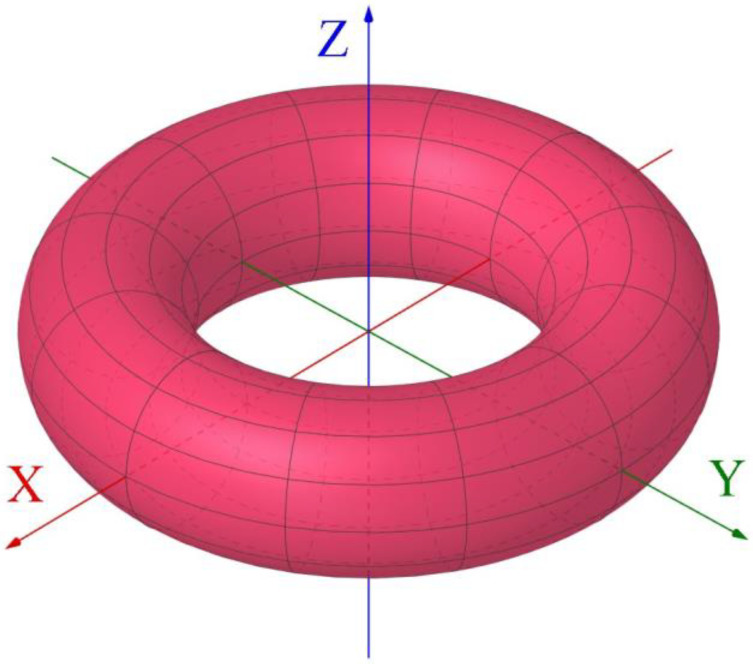
The schematic image of the toroidal magnetic hopfion and the coordinate system. The iso-surface of the hopfion magnetization component $m_{z}\left(\eta \right)=const$ or the toroidal parameter $\eta \left(x,y,z\right)=const$ is shown using red. The torus centers are located at the rings $\rho =a\;coth\left(\eta \right)$ in the *xOy*-plane. The scale parameter a is the hopfion radius.

**Figure 2 nanomaterials-14-00125-f002:**
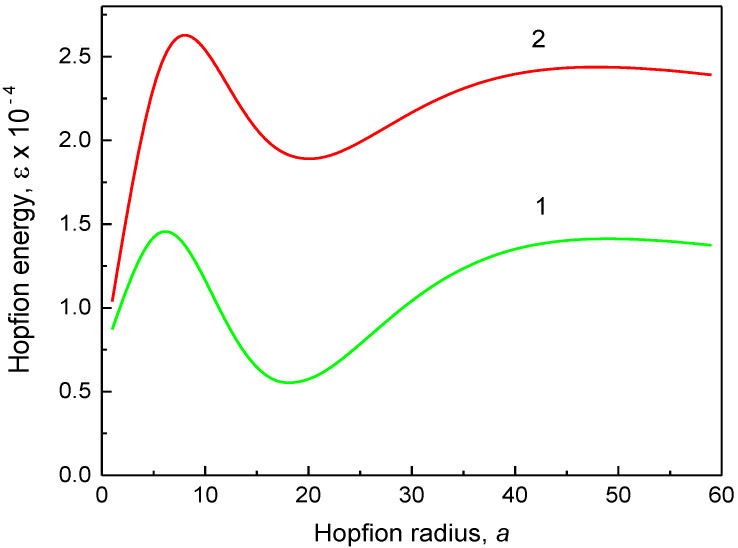
The hopfion energy $\varepsilon \left(a,R\right)$ vs. the hopfion radius, a (in units of the exchange length). The dot radius $R=20l_{e}$, the DMI parameter *d* = 1, and the exchange length $l_{e}=$ 5 nm. (1) A cylindrical dot with an aspect ratio of height/radius *β* = 2, and (2) a cylindrical wire with an aspect ratio of length/radius *β* = 8.

**Figure 3 nanomaterials-14-00125-f003:**
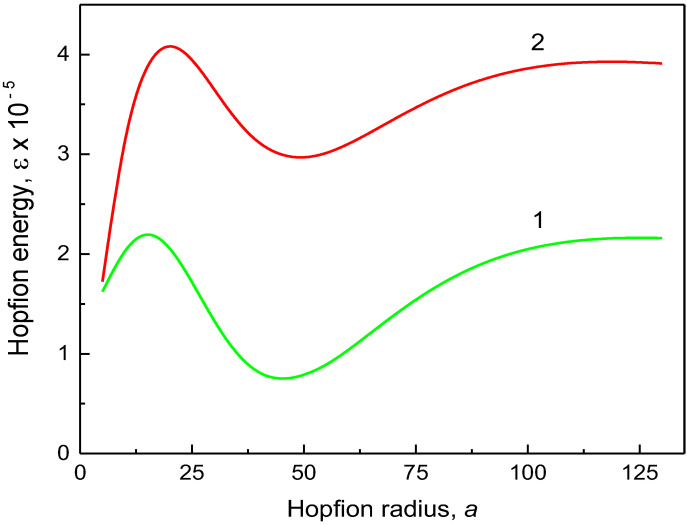
The hopfion energy $\varepsilon \left(a,R\right)$ vs. the hopfion radius, a (in units of the exchange length). The dot radius $R=50l_{e}$, the DMI parameter *d* = 1, and the exchange length $l_{e}=$ 5 nm. (1) A cylindrical dot with an aspect ratio of height/radius *β* = 2, and (2)a cylindrical wire with an aspect ratio of length/radius *β* = 10.

**Figure 4 nanomaterials-14-00125-f004:**
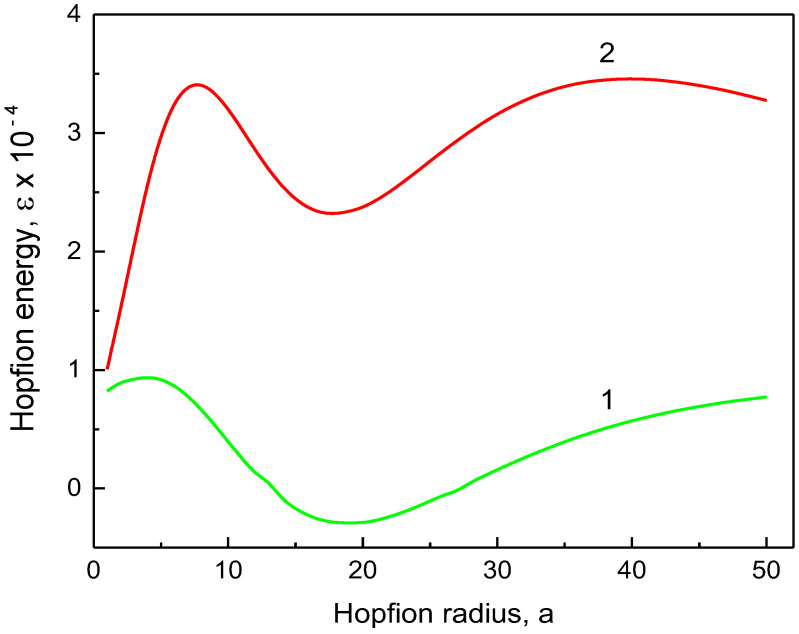
The hopfion energy $\varepsilon \left(a,R\right)$ vs. the hopfion radius, a (in units of the exchange length). The dot radius $R=20l_{e}$, the cylindrical dot with an aspect ratio of height/radius *β* = 2, the DMI parameter *d* = 1, and the exchange length $l_{e}=$ 5 nm. (1) “easy plane” magnetic anisotropy with $K/2\pi M_{s}^{2}=-0.5$; (2) “easy axis” magnetic anisotropy with $K/2\pi M_{s}^{2}=1$.

## Data Availability

Data are contained within the article.

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
