# Peer review of "3D Magnetization Textures: Toroidal Magnetic Hopfion Stability in Cylindrical Samples"

_nanomaterials, 2024, doi:10.3390/nano14010125_

Round 1
Reviewer 1 Report
Comments and Suggestions for Authors
The topic of this work is interesting, as apart from the possible applications topologically protected magnetic structures in information technology, it is always fascinating to find solid-state analogs of basic physics phenomena. The manuscript is well written, provides a thorough literature review and statement of the interest of the problem in the introduction and detailed description of the method. However, in its present form it seems to be addressing a narrow audience more specialized mathematical physics rather than the intended broad audience of chemistry of interfaces and of nanostructures and their applications.
To make more suitable to a broader audience I would propose the following:
· Include an image of a typical Hopfion spin-texture in the cylindrical nanostructure and the coordinates used.
· In the conclusions section, based on the micromagnetic material parameters given in the beginning of section 3, give some values of the calculated energy barriers in Kelvin.
As it regards the barrier calculation itself, I have some strong doubts: Is it based on rigidly inflating a specific spin structure to different values of the hopfion radius a? Is such a method sufficient? Shouldn’t some method as the “nudged elastic band” be used? For instance, with this method there is no discontinuity at the point a=R as one would intuitively expect. This point should be clarified and discussed.
Author Response
- The referee´s comment:
Include an image of a typical Hopfion spin-texture in the cylindrical nanostructure and the coordinates used.
My reply:
I plotted such hopfion image and inserted it to the revised manuscript along with brief comments (new Figure 1).
- The referee´s comment:
In the conclusions section, based on the micromagnetic material parameters given in the beginning of section 3, give some values of the calculated energy barriers in Kelvin.
My reply:
I inserted the typical values of the calculated energy barriers between the hopfion and out-of-plane single domain configurations in the end of Sec. 3 ¨Results and discussion¨.
- The referee´s comment/questions:
As it regards the barrier calculation itself, I have some strong doubts: Is it based on rigidly inflating a specific spin structure to different values of the hopfion radius a? Is such a method sufficient? Shouldn’t some method as the “nudged elastic band” be used? For instance, with this method there is no discontinuity at the point a=R as one would intuitively expect. This point should be clarified and discussed.
My reply:
Thank you for these valuable questions.
The energy barrier calculation is based on the ¨rigidly inflating¨ hopfion configuration by increasing or decreasing the hopfion radius, a. It is explicitly explained in the manuscript text.
The calculation method presented in the manuscript allows to conclude about the toroidal hopfion energy as a function of the hopfion radius and the energy minimum at some value of the hopfion radius a = a_0. I plotted in Figures 2-4 the toroidal hopfion energy vs. hopfion radius, a. Except a minimum at a = a_0, the hopfion energy vs. a reveal some maxima at a=a_m increasing and decreasing the value of a. The energy difference E(a_m) – E(a_0) can be treated as an energy barrier for the transitions from the hopfion to the out-of-plane single domain state (described as the limiting cases of zero and infinite values of a) within the toroidal hopfion model. However, the hopfion radius may be not appropriate ¨reaction coordinate¨ to describe a transition from the hopfion to the single domain magnetization configuration discussed in the manuscript. More realistic path in a multidimensional parameter space can involve some other intermediate magnetization configurations, which are very different from the toroidal magnetic hopfion. The saddle point along this path corresponds to an energy barrier which can be essentially smaller than the barrier calculated within the approach of the rigid hopfion transformation mode. An example of such intermediate magnetization configuration is ¨toron¨, see, for instance, the paper by Li et al., Phys. Rev. B 105, 174407 (2022), new Ref. 34.
The problem of the transformation of the hopfion to the out-of-plane single domain state (applying, for instance, a magnetic field along the axial symmetry Oz direction) is beyond scope of this manuscript. Some complicated numerical methods like the “nudged elastic band” can be used to find the corresponding energy barriers.
I revised the manuscript to clarify the situation with the energy barriers (see the last part of Sec. 3. Results and discussion).
Reviewer 2 Report
Comments and Suggestions for Authors
Here, K. Guslienko demonstrated that one of the topologically non-trivial magnetization configurations in ferromagnetic materials, called hopfion, is a metastable state for a thick cylindrical ferromagnetic nanodot or a nanowire of a finite radius. Author analyzed the energy and stability of the magnetic hopfion in magnetic cylindrical nanodots and nanowires, using the methods of the theory of micromagnetism and determine the main magnetic and geometrical parameters of existence of the stable hopfion configurations in the restricted cylindrical geometry.Then author reveals this 3D magnetization texture is a metastable state as a result of the competition of the exchange, magnetic anisotropy and magnetostatic energies, in which the latter is crucial for the toroidal hopfion stability. So, I think this is an enlightening manuscript and is suitable for publication in nanomaterials.
In addition, some advice as follows:
1, These non-trivial topologically magnetization configurations, like skyrmions, all have specific diagrams of magnetization texture. As a high-profile magnetic topological configuration, I suggest that the author should give an explicit magnetization texture of hopfion and compare it with other magnetization configurations, which I think the author should focus on in the introduction.
2, The author mentioned in the abstract and conclusion that the toroidal hopfion metastable magnetization configuration can be reached in the process of the sample re-magnetizing by applying an external magnetic field along the cylindrical axis. But there are no any specific details and reasons in the text, which will be of guiding significance to the actual experiments. So I suggest the author to check and add them.
3, A large number of formulas are involved in the text, and it is recommended that the author carefully check the writing and formatting problems. Moreover, authors should summarize some important parameters when determining whether a material structure has a hopfin magnetization texture. And as for the materials with hopfin magnetization configuration, the author only raises Ni80Fe20, which I think is far from enough. So I suggest the author to give more practical cases.
4,I believe it is necessary to review the potential applications of this unique toroidal hopfion magnetic configuration in the part of introduction and suggest the author refer to the following two literatures on toroidal topological molecules, one is DOI:10.1021/jacs.2c05421 and the other is DOI:10.1016/j.matt.2020.02.021, which described the molecule-based nanomagnets with toroidal arrangement of metal ions can not only generate ferrotoroidictiy but also regulate spin wave excitations. Maybe there is a lot of similarities in these toroidal magnetic materials, which all have attracted considerable attention in nano-fileds.
Author Response
- The referee´s comment:
These non-trivial topologically magnetization configurations, like skyrmions, all have specific diagrams of magnetization texture. As a high-profile magnetic topological configuration, I suggest that the author should give an explicit magnetization texture of hopfion and compare it with other magnetization configurations, which I think the author should focus on in the introduction.
My reply:
The explicit hopfion magnetization texture is presented by Equations (3) and (3´) in the manuscript. My manuscript is about some particular 3D magnetization texture such as the toroidal hopfion. This is not a review paper. Therefore, I cannot focus on other magnetization textures due to the limited length of manuscript. I briefly mentioned in the manuscript other magnetization textures: vortex and skyrmion (2D), Bloch point and toron (3D).
- The referee´s comment:
The author mentioned in the abstract and conclusion that the toroidal hopfion metastable magnetization configuration can be reached in the process of the sample re-magnetizing by applying an external magnetic field along the cylindrical axis. But there are no any specific details and reasons in the text, which will be of guiding significance to the actual experiments. So I suggest the author to check and add them.
My reply:
My manuscript is theoretical research about the toroidal magnetic hopfion stability in the cylindrical geometry. I did not include the Zeeman energy (magnetic field term) to the energy functional Equation (12) and did not consider its influence on the hopfion stability. I only mentioned in the end of Sec. 3 that the hopfion has a finite remanent magnetization along axial direction Oz. The sample magnetization hysteresis loop applying an external magnetic field in any direction is a consequence of the metastable magnetization configurations. One of these configurations may be the toroidal hopfion configuration. This is just a theoretical opportunity and, therefore, I cannot supply any guidance for experimental observation of the toroidal hopfions by measuring the hysteresis loops.
I inserted two sentences about the magnetic hopfion hysteresis loop in the end of Sec. 3.
- The referee´s comment/questions:
A large number of formulas are involved in the text, and it is recommended that the author carefully check the writing and formatting problems. Moreover, authors should summarize some important parameters when determining whether a material structure has a hopfin magnetization texture. And as for the materials with hopfion magnetization configuration, the author only raises Ni80Fe20, which I think is far from enough. So I suggest the author to give more practical cases.
My reply:
I checked all equations included to the manuscript. The material and geometrical parameters important for the toroidal hopfion stabilization are underlined in Abstract, the manuscript text (Secs. 2, 3) and Conclusions.
The particular choice of the magnetic parameters of pemalloy (Ni80Fe20) for numerical calculation of the hopfion energy does not influence the main results of the manuscript presented in Figures 2-4. The reduced hopfion magnetic energy (12) is valid for any magnetic material with the uniaxial magnetic anisotropy (the parameter Q) and Dzyaloshinskii-Moria exchange interaction (the parameter d) for the given cylindrical sample sizes. The material exchange length l_e serves as a natural scale for the hopfion radius.
I revised the beginning of Sec. 3 to underline the universal character of the reduced hopfion energy given by Equation (12).
- The referee´s comment/questions:
I believe it is necessary to review the potential applications of this unique toroidal hopfion magnetic configuration in the part of introduction and suggest the author refer to the following two literatures on toroidal topological molecules, one is DOI:10.1021/jacs.2c05421 and the other is DOI:10.1016/j.matt.2020.02.021, which described the molecule-based nanomagnets with toroidal arrangement of metal ions can not only generate ferrotoroidictiy but also regulate spin wave excitations. Maybe there is a lot of similarities in these toroidal magnetic materials, which all have attracted considerable attention in nano-fileds.
My reply:
I briefly mentioned the potential applications of the toroidal magnetic hopfions in Introduction.
I overviewed both papers on the toroidal topological molecules, H.-L. Zhang et al., {ScnGdn} Heterometallic Rings: Tunable Ring Topology for Spin-Wave Excitations. J. Am. Chem. Soc. 2022, 144, 33, 15193–15202 and H.-L. Zhang et al., Single-Molecule Toroic Design through Magnetic Exchange Coupling, Matter 2, 1481–1493 (2020). Although some interesting results were presented there, the contents of these papers are very far from the presented manuscript, where the 3D magnetic solitons (toroidal hopfions) were investigated in relatively large ferromagnetic cylindrical dots.
Reviewer 3 Report
Comments and Suggestions for Authors
Topological magnetization defects are currently attracting interest due to a number of applied advantages. An important feature of such configurations is their stability, so this problem is being intensively studied. Defects such as skyrmions, vortices and domain walls have been well studied. Hopfions, which are a 3D defect, are of no less interest, but have been studied much less well. In this work, a theoretical micromagnetic study of the stability of a hopfion confined within a cylindrical body was carried out. As a result of the theoretical analysis performed, the author established that the localization of hopfion in the cylinder is characterized by a minimum of energy, leading to a certain equilibrium size of the latter. The author argues that the main energies in this case are exchange energy, magnetostatic energy and magnetic anisotropy energy. It is argued that the Dzyaloshinsky-Moriya interaction and the energy of surface magnetic anisotropy are secondary. These results are of undoubted interest to readers of the journal, since a new phenomenon at the nanoscale is predicted. The material is generally suitable for Nanomaterials, but I would like to clarify a couple of points.
1. The toroidal configuration assumes that its description requires 2 parameters: the large and small radius of the torus. Why can a minimum energy observed in only one of the parameters be considered as evidence of stability?
2. The text does not directly explain what specific size the cylinder in which the hopfion is stable should be. Are these sizes related to the nanoscale, or does the size of the cylinder not matter at all?
3. Considering the serious experimental activity on magnetic dots and nanowires, including observations of their magnetic microstructure, one would expect that such a texture should already be observed in the experiment. What do you think is the reason that this is not so?
Author Response
- The referee´s comment:
The toroidal configuration assumes that its description requires 2 parameters: the large and small radius of the torus. Why can a minimum energy observed in only one of the parameters be considered as evidence of stability?
My reply:
There is only one geometrical parameter to describe the toroidal magnetic hopfion, the hopfion radius, ¨a¨. The out-of-plane hopfion magnetization component, m_z is function only of the toroidal parameter η. The iso-surfaces of the constant magnetization component m_z (or constant toroidal parameter η) are non-intersecting tori, see new Figure 1. The torus centers are located at the rings ρ=a*coth(η) in the xy-plane and the torus radii are equal to a/sinh(η). Therefore, the ¨large radius of the torus¨ is equal to a*coth(η), and ¨small radius of the torus¨ is equal to a/sinh(η). See more details after Eq. (12) of Ref. 27 [Guslienko, Chaos 2023].
- The referee´s comment:
The text does not directly explain what specific size the cylinder in which the hopfion is stable should be. Are these sizes related to the nanoscale, or does the size of the cylinder not matter at all?
My reply:
The cylinder size is important for the hopfion stabilization. The cylinder radius R and cylinder height L in the manuscript are taken in the units of the material exchange length l_e = 5 nm. To plot Figures 2-4 in the manuscript I did use the cylinder radii R=20*l_e = 100 nm, R=50*l_e = 250 nm, and the cylinder aspect ratios beta = L/R equal to 2, 8 and 10. I also checked existence of the minimum of the cylinder magnetic energy as a function of the hopfion radius for R=5*l_e =25 nm, R=10*l_e= 50 nm, and R=100*l_e= 500 nm.
I specified the cylinder sizes in absolute units (nm) in the revised manuscript (Sec. 3).
- The referee´s comment:
Considering the serious experimental activity on magnetic dots and nanowires, including observations of their magnetic microstructure, one would expect that such a texture should already be observed in the experiment. What do you think is the reason that this is not so?
My reply:
The magnetic hopfions were observed experimentally by using magnetic transmission soft X-ray microscopy and X-ray photoemission electron microscopy in the thin multilayer films Ir/Co/Pt shaped to nanoscale disks, see Ref. 4 [Kent et al., Nat. Commun. 2021] of the present manuscript.
Very recently (Zheng, F., Kiselev, N.S., Rybakov, F.N. et al. Hopfion rings in a cubic chiral magnet. Nature 623, 718–723 (2023). https://doi.org/10.1038/s41586-023-06658-5, published on Nov. 22, 2023) the hopfions (hopfion rings) were observed in FeGe plates using Lorentz imaging and electron holography. The reference to the paper by Zheng et al. (Nature 2023) was included to the revised manuscript as new Ref. 5.
Round 2
Reviewer 1 Report
Comments and Suggestions for Authors
After the amendments the manuscript is suitable for publication in its present form.